# Research on Automatic Generation Control with Wind Power Participation Based on Predictive Optimal 2-Degree-of-Freedom PID Strategy for Multi-area Interconnected Power System

**Xilin Zhao \*** , **Zhenyu Lin** , **Bo Fu \*** , **Li He and Na Fang**

Hubei Key Laboratory for High-efficiency Utilization of Solar Energy and Operation Control of Energy Storage System, Hubei University of Technology, Wuhan 430068, China; 101710324@hbut.edu.cn (Z.L.); heli@hbut.edu.cn (L.H.); 20050009@hbut.edu.cn (N.F.)

\* Correspondence: zhaoxl@mail.hbut.edu.cn (X.Z.); fubofanxx@mail.hbut.edu.cn (B.F.); Tel.: +86-15607181737 (X.Z.); +86-27-5975-0433 (B.F.)

**Abstract:** High penetration of wind power in the modern power system renders traditional automatic generation control (AGC) methods more challenging, due to the uncertainty of the external environment, less reserve power, and small inertia constant of the power system. An improved AGC method named predictive optimal 2-degree-of-freedom proportion integral differential (PO-2-DOF-PID) is proposed in this paper, which wind farm will participate in the load frequency control process. Firstly, the mathematical model of the AGC system of multi-area power grid with penetration of wind power is built. Then, predictive optimal 2-degree-of-freedom PID controller is presented to improve the system robustness considering system uncertainties. The objective function is designed based on the wind speed and whether wind farm takes part in AGC or not. The controller solves the optimization problem through the predictive theory while taking into account given constraints. In order to obtain the predictive sequence of output of the whole system, the characteristic of the 2-DOF-PID controller is integrated in the system model. A three interconnected power system is introduced as an example to test the feasibility and effectiveness of the proposed method. When considering the penetration of wind power, two cases of high wind speed and low wind speed are analyzed. The simulation results indicate that the proposed method can effectively deal with the negative influence caused by wind power when wind power participates in AGC.

**Keywords:** wind power; automatic generation control; predictive optimality; 2-degree-of-freedom PID

---

## 1. Introduction

In modern power systems, automatic generation control (AGC) is used as the foundation of secondary frequency control to maintain the frequency of the power system close to its scheduled value [1]. Because of the negative impact of traditional coal and fossil fuel-based power generation to the environment, renewable energy such as wind power and photovoltaic has received fast growing attention throughout the world and the utilization of such energy has increased remarkably over the past decades [2,3]. However, the rapidly increase in the penetration level of renewable energy is challenging the traditional way of AGC due to the decrease of generation units providing reserve power for AGC and reduction of the inertia of the whole power system [4]. In order to meet the challenge, the research on the advanced control scheme which wind farm participate in the load frequency control process of the system can serve as a counter measure [5].

Generally, proportion integral differential (PID) controller is still the most popular method applied to AGC for its practicability [6]. The parameters of conventional PID controller are fixed, which are not suitable for the complexity and nonlinear characteristic of power system. Therefore, some research, such as 2-degree-of-freedom proportional integral derivative (2-DOF PID) [7], fractional order PID [8], and proportional integral derivative plus second order derivative [9], demonstrated a commitment to adjust the structure of the controller to improve control performance. When considering the penetration of renewable energy, the adjustment of the controller structure is inadequately to compensate the disadvantage caused by the uncertainty of renewable generation [10]. Thus, some optimal control methods are used to deal with the problem. Mcnamara et al. proposed model predictive control (MPC) as a means to implement automatic generation control for power system [11]. Xu et al. researched a dynamic gain-tuning control method to adjust the parameters of PID controller online during the AGC process [12]. Arya et al. presented an output scaling factor based fuzzy controller to enrich AGC conduct of two-area electrical power systems and employ integral of squared error criterion to optimize the output scaling factor (SF) of fuzzy proportional integral controller [13]. Tummala et al. presented a novel sliding mode controller with non-linear disturbance observer to effectively mitigate the wide changes in frequency [14].

To some extent, existing methods can show good control performance on the balance between generation and load concerning the complexity and uncertainty for modern power systems [15]. However, the problem will be complicated when power system has high penetration of wind power. The complexity is manifested in whether the wind power takes part in the frequency regulation or not [16]. Generally, the increased penetration of wind power introduces challenges in not only the extra power fluctuation caused by the uncertainty of wind, but also the reduction of synchronous inertia due to the electrical decoupling with the grid by a power electronic converter [17]. Therefore, when wind power does not take part in the frequency regulation, the wind power fluctuation can be equivalent to a form of load disturbance, and the reserve capacity of the system needs to be increased to provide sufficient capacity for damp power imbalances [18]. On the other hand, with respect to the controllability of wind power units, wind generation can take part in the frequency regulation, which will maintain the reserve capacity of the system adequately [19].

At present, there are two modes that wind power takes part in the frequency regulation. One is the exchange management of kinetic energy stored in the blades and generator to offer grid support [4]. Wang et al. investigated the implementation of inertial response and primary frequency control in a wind turbine controller [19]. Wickramasinghe et al. proposed a method to temporarily convert doubly-fed induction generators (DFIG) to synchronous generators, enabling supply of real inertia to the system, instead of emulating inertia [20]. Fu et al. proposed a novel integrated frequency governor applied to a wind turbine to provide fast active power support and scheduled power allocation for both temporary inertial response and continued primary frequency regulation [21]. However, in the process of inertial response, power released by the rotor of wind turbine is equal to the power absorbed, which cannot provide additional energy to the power system in the long term. Therefore, the inertial response process is short-term and second frequency drop may happen [4].

The other mode is a flexibility method for wind power units to provide frequency regulation, which the reference point of the wind power units is adjusted according to the equilibrium state of power generation and load [22]. Compared with the virtual inertial control, the deloading control by adjusting the pitch angle can provide a wider range of regulation and long-term frequency support [23]. Civelek et al. recommend a new fuzzy logic proportional control approach in order to mitigate the moment load on blades and tower to a minimum possible value while keeping the output power of WTs at a constant value [24]. Lio et al. presented a controller enables clear and transparent quantification of the benefits gained by using preview control [25]. Liu et al. proposed a switching angle controller and an automatic generation controller for the DFIG to control the frequency of DFIG-based wind power penetrated power systems [16].

In this paper, an improved predictive optimal 2-degree-of-freedom PID (PO-2-DOF-PID) controller is proposed for AGC of power system with high penetration of wind power. The main purpose of the controller design is to pursue better control performance when wind power units take part in the AGC process. Because the parameters of traditional PID controller are fixed, the characteristic cannot reply the uncertainty of the system well [26]. Meanwhile, owing to large number of iterations and long search time, intelligent optimization algorithm is not conducive to online implementation. To solve the problem, a predictive optimal unit is cascaded with the 2-degree-of-freedom PID controller. The input of the controller can be adjusted by the predictive optimal result to enhance the robustness of the system and obtain better performance. In addition, the control object of predictive controller is the system model with 2-DOF-PID controller, that means the combination of 2-DOF-PID and predictive unit does not increase the complexity of the control system and is easy to implement.

This paper is organized as follows. After introducing the background of the research, the dynamic model of concerned system with high penetration of wind power is described in Section 2, which includes the dynamic model of wind turbine. In Section 3, predictive optimal algorithm is presented, together with the description of PO-2-DOF-PID controller and the mode about how wind power take part in the AGC process. In Section 4, a three-area connected power system is discussed as a numerical example. Two cases, which represent the high speed and low speed of the wind, are analyzed to illustrate the effectiveness of the proposed method. Finally, the conclusion is presented in Section 5.

## 2. Model Description

### 2.1. Distributed Model of Interconnected Power System

For the geographical disparity, modern power system is characterized by multi-area interconnected form. The block diagram of the interconnected system is illustrated in Figure 1, as a distributed AGC mode. The controller in each area exchanges information through a communication channel. Additionally, there is a power transmission line between each area for electric energy transmitting. Without loss of generality, a three-area interconnected power system is analyzed in this paper. In area 1, there is wind power and thermal power, both participating in frequency adjustment of the power system. In area 2, there are wind power and thermal power like area 1, but thermal power is responsible for the frequency modulation, while wind power runs in the MPPT state to deliver power to the grid, and does not participate in frequency regulation. In area 3, there are only thermal powers. During the research process, assuming that there is no time delay between areas, and all the wind turbines are equivalent to one turbine.

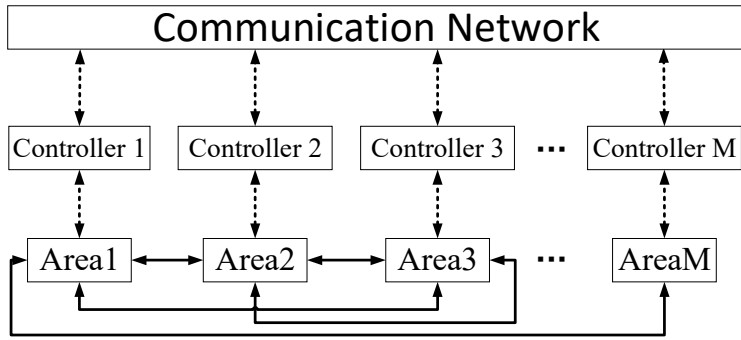

**Figure 1.** Multi-area interconnected power system.

### 2.2. AGC Model with Thermal Power

Thermal power generator is the conventional AGC unit. The block diagram of a typical AGC system with reheat thermal power plant is shown in Figure 2. Based on the principle of system equivalence and simplification, the system consists of speed governing subsystem (SG), reheat time

delay subsystem (RTD), steam turbine unit (STU) and power system (PS). The mathematic model of these units can be described as follows [27]:

$$\text{STU}: \Delta \dot{P}_{gi} = -\frac{1}{T_{ti}}\Delta P_{gi} - \frac{1}{T_{ti}}\Delta P_{ri}, \tag{1}$$

$$\text{SG}: \Delta \dot{X}_{gi} = -\frac{1}{T_{gi}R_i}\Delta f_i - \frac{1}{T_{gi}}\Delta X_{gi}, \tag{2}$$

$$\text{RTD}: \Delta \dot{P}_{ri} = -\frac{K_{ri}}{T_{gi}R_i}\Delta f_i + (\frac{1}{T_{ri}} - \frac{K_{ri}}{T_{gi}})\Delta X_{gi} - \frac{1}{T_{ri}}\Delta P_{ri}. \tag{3}$$

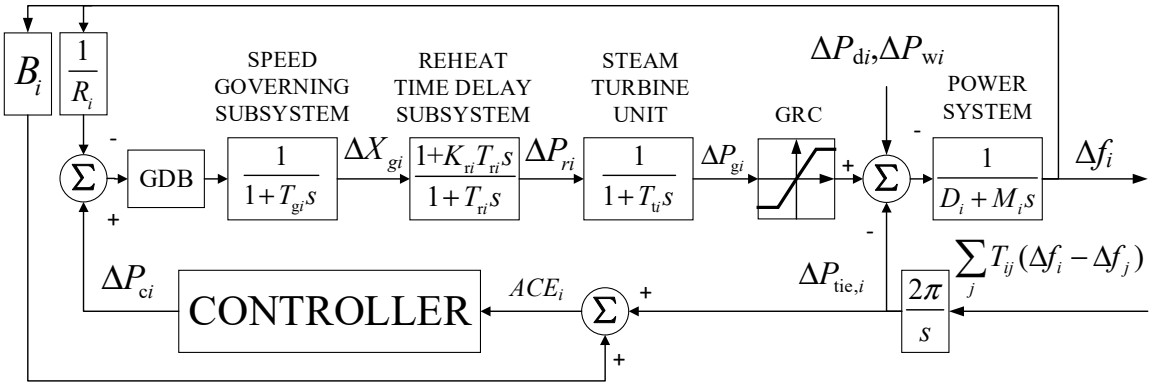

**Figure 2.** Block diagram of AGC system with thermal power.

During the AGC process, the AGC units are adjusted to maintain the stable of system frequency and tie-line flows. According to the block diagram of the system, the frequency deviation $\Delta f_i$ of area $i$ ($i = 1,2, \ldots , M$) is given by

$$\Delta \dot{f}_i = -\frac{D_i}{M_i}\Delta f_i + \frac{1}{M_i}(\Delta P_{gi} - \Delta P_{\text{tie},i} - \Delta P_{di} - \Delta P_{wi}) + \frac{1}{T_{gi}}\Delta P_{ci}. \tag{4}$$

The tie-line power flow change between areas $i$ and $j$ is given by

$$\Delta \dot{P}_{\text{tie}}^{ij} = T_{ij}(\Delta f_i - \Delta f_j), \Delta P_{\text{tie}}^{ij} = -\Delta P_{\text{tie}}^{ij}. \tag{5}$$

The total tie-line power change between different areas can be derived as

$$\Delta \dot{P}_{\text{tie},i}^{ij} = \sum_{\substack{j=1 \\ j \neq i}}^{3} \Delta \dot{P}_{tie}^{ij} = \sum_{\substack{j=1 \\ j \neq i}}^{3} T_{ij}(\Delta f_i - \Delta f_j). \tag{6}$$

The area control error (ACE) is determined by

$$ACE_i = B_i + \Delta P_{\text{tie},ij}. \tag{7}$$

The parameters of the AGC system are shown in Table 1.

**Table 1.** Automatic generation control (AGC) system parameters description.

| Parameter/Variable | Description | Unit |
|---|---|---|
| $\Delta f_i(t)$ | Frequency deviation | Hz |
| $\Delta P_{gi}(t)$ | Generator output power deviation | p.u. |
| $\Delta X_{gi}(t)$ | Governor valve position deviation | p.u. |
| $\Delta P_{\text{tie},i}(t)$ | Tie-line active power deviation | p.u. |
| $\Delta P_{di}(t)$ | Load disturbance | p.u. |
| $\Delta P_{wi}(t)$ | Wind power disturbance | p.u. |
| $M_i$ | Generator moment of inertia | kg·m$^2$ |
| $K_{ri}$ | Reheat turbine gain | Hz/p.u. |
| $D_i$ | Damping constant for area $i$ | s |
| $T_{ri}$ | Reheat turbine time constant | s |
| $T_{gi}$ | Thermal governor time constant | s |
| $T_{ti}$ | Turbine time constants | s |
| $T_{ij}$ | Interconnection gain between control areas | p.u. |
| $B_i$ | Frequency bias factor | p.u./Hz |
| $R_i$ | Speed drop due to governor action | Hz/p.u. |
| $ACE_i$ | Area control error | p.u. |

Form Equations (1)–(7), the state model of the system of area $i$ can be derived as follows

$$\begin{cases} \dot{X} = AX + Bu + D\omega \\ y = CX \end{cases}, \tag{8}$$

where $X = \begin{bmatrix} \Delta f_i & \Delta P_{\text{tie},i} & \Delta P_{gi} & \Delta X_{gi} & \Delta P_{ri} \end{bmatrix}^{\text{T}}$, $u = \begin{bmatrix} \Delta P_{ci} \end{bmatrix}$, $\omega = \begin{bmatrix} \Delta P_{di} & \Delta P_{wi} \end{bmatrix}^{\text{T}}$,

$$A = \begin{bmatrix} -\frac{D_i}{M_i} & -\frac{1}{M_i} & -\frac{1}{M_i} & 0 & 0 \\ T_{ij} & 0 & 0 & 0 & 0 \\ 0 & 0 & -\frac{1}{T_{ti}} & 0 & \frac{1}{T_{ti}} \\ -\frac{1}{T_{gi}R_i} & 0 & 0 & -\frac{1}{T_{gi}} & 0 \\ -\frac{K_{ri}}{T_{gi}R_i} & 0 & 0 & \frac{1}{T_{ri}} - \frac{K_{ri}}{T_{gi}} & -\frac{1}{T_{ri}} \end{bmatrix}, \quad B = \begin{bmatrix} 0 \\ 0 \\ 0 \\ \frac{1}{T_{gi}} \\ 0 \end{bmatrix}, \quad D = \begin{bmatrix} -\frac{1}{M_i} & -\frac{1}{M_i} \\ 0 & 0 \\ 0 & 0 \\ 0 & 0 \\ 0 & 0 \end{bmatrix}, \quad C = \begin{bmatrix} B_i \\ 1 \\ 0 \\ 0 \\ 0 \end{bmatrix}.$$

### 2.3. AGC Model with Wind Farm Participation

The block diagram of the AGC system which wind power take part in the frequency regulation is shown in Figure 3.

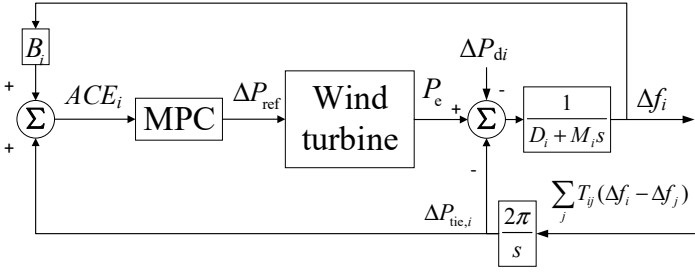

**Figure 3.** Block diagram of AGC system with wind power participation.

When wind generations take part in the frequency regulation, the model of the wind turbine need to be analyzed. According to the aerodynamic principle of the wind turbine, the torque of wind turbine is described as [28]

$$T_{\text{a}} = \frac{0.5\pi\rho R^2 V_{\text{m}}^3 C_{\text{p}}(w_{\text{r}}, \theta)}{w_{\text{r}}}, \tag{9}$$

where $\rho$ is the air density, $R$ is the blade length, $V_m$ is the wind speed, $w_r$ is the speed of the rotor, $C_p$ is the power coefficient which is a function of blades pitch angle $\theta$ and tip-speed ratio $\lambda$. In addition, the first-order derivative of $w_r$ is given by (10), and the relationship of the speed between rotor and generator of the wind turbine can be obtained as (11).

$$\dot{w}_r = \frac{1}{J_t}(T_a - N_g T_g), \tag{10}$$

$$w_g = N_g w_r \tag{11}$$

where $T_g$ is the electrical torque, $w_g$ is the speed of the generator, $N_g$ is the gear box ratio, $J_r$ is the rotor inertia, $J_g$ is the generator inertia, and $J_t = J_r + N_g^2 J_g$.

Then, the relationship between the torque and the output of the wind turbine is given by

$$T_{g,ref} = \frac{P_{ref}}{w_g}. \tag{12}$$

Generally, because the vector control is applied in the local torque control loop, ensuring a fast and accurate response, we have an equivalent as follows [29]:

$$T_{g,ref} \approx T_g, \ P_e = \mu P_{ref}, \tag{13}$$

where $\mu$ denotes the generator efficiency, which is 94.4% in this paper [29].

In addition, the output of wind power can be adjusted through the pitch angle of wind turbine $\theta$ which is determined by

$$\theta = -\frac{K_p}{K_c}\dot{w}_f - \frac{K_i}{K_c}(w_f - w_{ref}), \tag{14}$$

where, $P_{ref}$ is wind farm power reference, $P_e$ is the wind farm output power, $w_f$ is the deviation of the filtered generator speed, $w_{ref}$ is the rated speed, $K_c$ is the correction factor, $K_p$ and $K_i$ are proportional and integral gain of the PI controller which is integrated in the wind turbine by manufacturer.

When considering the combined power supply of wind power and thermal power, the state model of the system can be derived as [30].

$$X = \begin{bmatrix} w_r & w_f & \theta & \Delta f_i & \Delta P_{tie,i} & \Delta P_e \end{bmatrix}^T, \ u = [\Delta P_{ref}], \ \omega = \begin{bmatrix} V_m & \Delta P_{di} \end{bmatrix}^T,$$

$$A = \begin{bmatrix} \frac{1-N_g}{J_t} & 0 & \frac{1}{J_t} & 0 & 0 & 0 \\ \frac{N_g}{J_t} & \frac{1}{T_g} & 0 & 0 & 0 & 0 \\ -\frac{K_p N_g}{K_c T_g} & \frac{K_p - K_i T_g}{K_c T_g} & 0 & 0 & 0 & 0 \\ 0 & 0 & 0 & -\frac{D_i}{M_i} & -\frac{1}{M_i} & \frac{1}{M_i} \\ 0 & 0 & 0 & T_{ij} & 0 & 0 \\ 0 & 0 & 0 & 0 & 0 & 0 \end{bmatrix}, \ B = \begin{bmatrix} -\frac{1}{J_t} \\ 0 \\ 0 \\ 0 \\ 0 \\ 0 \end{bmatrix}, \ D = \begin{bmatrix} -\frac{N_g}{J_t} & 0 \\ 0 & 0 \\ 0 & 0 \\ 0 & -\frac{1}{M_i} \\ 0 & 0 \\ 0 & 0 \end{bmatrix}, \ C = \begin{bmatrix} 0 \\ 0 \\ 0 \\ B_i \\ 1 \\ 0 \end{bmatrix}.$$

### 2.4. Nonlinear Constraint Processing

Considering the physical characteristics of each unit, some constraints such as valve position, generation rate constraint (GRC) and governor dead band (GDB) need to be considered in the analysis of AGC.

The GRC in the thermal power is expressed as $\mu_{min} \leq \Delta P_{gi}(k) \leq \mu_{max}$, where $\mu_{min}$, $\mu_{max}$ are set as $-0.0017$, $0.0017$ in the verification process respectively.

Refer to reference [31], the GDB is represented as

$$GDB = 0.8x_i - \frac{0.2}{\pi}x_i, \tag{15}$$

where $x_i$ is the state-variable and $i$ stands for the serial number of area.

## 3. Predictive Optimal 2-DOF PID Control Strategy

### 3.1. Structure of the Controller

In order to enhance the robustness and control performance of the system, a predictive optimal 2-DOF-PID controller is proposed. The controller is characterized by a cascade structure where the output of the predictive optimal unit is connected to the input of a 2-DOF-PID controller. Thus, the input of the PID controller can be adjusted by the predictive optimal result to meet the uncertainty of the system and to obtain better control performance. The structure of a 2-DOF-PID controller is shown in Figure 4a. When controller is designed as a PO-2-DOF-PID, the block diagram of the AGC system where the AGC units are wind power and thermal power are shown in Figure 4b,c respectively.

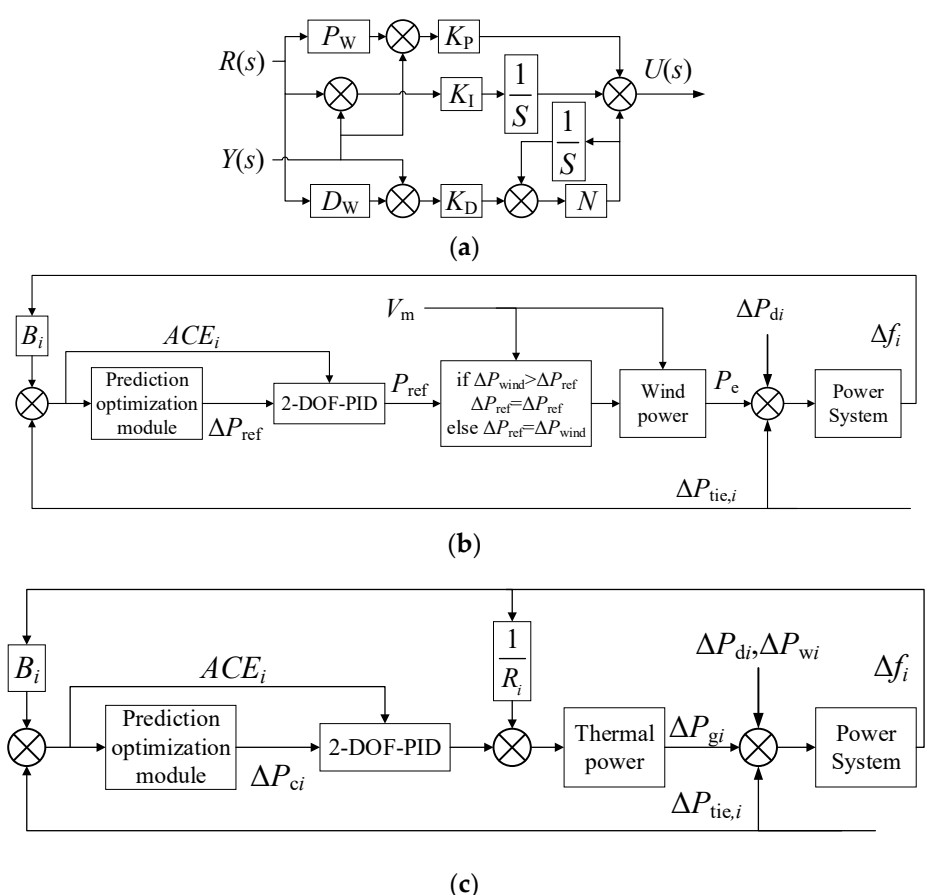

**Figure 4.** Block diagram of AGC system based on PO-2-DOF-PID controller. (**a**) Conventional 2-DOF-PID controller. (**b**) AGC system with wind power. (**c**) AGC system with thermal power.

From Figure 4a, the transfer function of the 2-DOF PID is given by

$$U(s) = K_{\text{P}i}(P_{\text{W}} \times R(s) - Y(s)) + K_{\text{I}i}\frac{1}{s}(R(s) - Y(s)) + K_{\text{D}i}\frac{N}{1 + N\frac{1}{s}}(D_{\text{W}} \times R(s) - Y(s)). \quad (16)$$

When the AGC unit is thermal power, the mathematic model of the system with a 2-DOF-PID controller is given by

$$\boldsymbol{X} = \begin{bmatrix} \Delta f_i & \Delta P_{\text{tie},i} & \Delta P_{gi} & \Delta X_{gi} & \Delta P_{ri} & \Delta P_{\text{I}i} & \Delta P_{\text{D}i} \end{bmatrix}^{\text{T}}, \boldsymbol{u} = [\Delta P_{ci}], \boldsymbol{\omega} = \begin{bmatrix} \Delta P_{di} & \Delta P_{wi} \end{bmatrix}^{\text{T}},$$

$$A = \begin{bmatrix}
-\frac{D_i}{M_i} & -\frac{1}{M_i} & \frac{1}{M_i} & 0 & 0 & 0 & 0 \\
T_{ij} & 0 & 0 & 0 & 0 & 0 & 0 \\
0 & 0 & -\frac{1}{T_{ti}} & \frac{K_{ri}}{T_{ti}} & \frac{1}{T_{ti}} & 0 & 0 \\
-\frac{\frac{1}{K_i}+B_i(NK_{Di}+K_{Pi})}{T_{gi}} & -\frac{NK_{Di}+K_{Pi}}{T_{gi}} & 0 & -\frac{1}{T_{gi}} & 0 & \frac{1}{T_{gi}} & \frac{1}{T_{gi}} \\
0 & 0 & 0 & \frac{1-K_{ri}}{T_{ri}} & \frac{1}{T_{ri}} & 0 & 0 \\
-B_i K_{Ii} & -K_{Ii} & 0 & 0 & 0 & 0 & 0 \\
-B_i K_{Di} N^2 & -K_{Di} N & 0 & 0 & 0 & 0 & -N
\end{bmatrix}, B = \begin{bmatrix} 0 \\ 0 \\ 0 \\ \frac{P_W K_{Pi}+D_W NK_{Di}}{T_{gi}} \\ 0 \\ K_{Ii} \\ -D_W N^2 K_{Di} \end{bmatrix}, D = \begin{bmatrix} -\frac{1}{M_i} & -\frac{1}{M_i} \\ 0 & 0 \\ 0 & 0 \\ 0 & 0 \\ 0 & 0 \\ 0 & 0 \\ 0 & 0 \end{bmatrix}.$$

Considering the participation of wind power, the mathematic model of the AGC system with the 2-DOF-PID controller can be derived as follows:

$$X = \begin{bmatrix} w_r & w_f & \theta & \Delta f_i & \Delta P_{tie,i} & \Delta P_e & \Delta P_{Ii} & \Delta P_{Di} \end{bmatrix}^T, u = [\Delta P_{ref}], \omega = \begin{bmatrix} V_m & \Delta P_{di} \end{bmatrix}^T,$$

$$A = \begin{bmatrix}
\frac{1-N_g}{J_t} & 0 & \frac{1}{J_t} & -\frac{B_i(K_{Di}N+K_{Pi})}{J_t} & -\frac{K_{Di}N+K_{Pi}}{J_t} & 0 & -\frac{1}{J_t} & -\frac{1}{J_t} \\
\frac{N_g}{J_t} & \frac{1}{T_g} & 0 & 0 & 0 & 0 & 0 & 0 \\
-\frac{K_P N_g}{K_c T_g} & \frac{K_P - \dot{K}_i T_g}{K_c T_g} & 0 & 0 & 0 & 0 & 0 & 0 \\
0 & 0 & 0 & -\frac{D_i}{M_i} & -\frac{1}{M_i} & \frac{1}{M_i} & 0 & 0 \\
0 & 0 & 0 & T_{ij} & 0 & 0 & 0 & 0 \\
0 & 0 & 0 & 0 & 0 & 0 & 0 & 0 \\
0 & 0 & 0 & -B_i K_{Ii} & -K_{Ii} & 0 & 0 & 0 \\
0 & 0 & 0 & -B_i K_{Di} N^2 & -K_{Di} N & 0 & 0 & -N
\end{bmatrix}, B = \begin{bmatrix} -\frac{P_W K_{Pi}+D_W NK_{Di}}{J_t} \\ 0 \\ 0 \\ 0 \\ 0 \\ 0 \\ K_{Ii} \\ D_W N^2 K_{Di} \end{bmatrix}, D = \begin{bmatrix} -\frac{N_g}{J_t} & 0 \\ 0 & 0 \\ 0 & 0 \\ 0 & -\frac{1}{M_i} \\ 0 & 0 \\ 0 & 0 \\ 0 & 0 \\ 0 & 0 \end{bmatrix}.$$

### 3.2. Predictive Control Problem Formulation

When the mathematic model of the system which includes the 2-DOF-PID controller is derived as aforementioned, predictive unit will adjust the input of the 2-DOF-PID controller to optimize the AGC performance through the obtainment of predictive sequence and the design of the objective function.

In general, if the sampling instant is set as $T_s$, the state equation of the discrete system can be written as followings:

$$x_i(k+1) = A_i(k)x_i(k) + B_i(k)u_i(k) + D_i(k)\omega_i(k) + \sum_{\substack{j \neq i \\ j=1}} (A_{ij}(k)x_j(k) + B_{ij}(k)u_j(k))$$

$$y_i(k) = C_i x_i(k)$$

(17)

In (17), $A_i$, $B_i$, $C_i$, $D_i$ represent appropriate system matrices for the control area $i$. $A_{ij}$, $B_{ij}$ represent the interaction matrices between area $i$ and area $j$.

The AGC system maintains the stability of the power system frequency by adjusting the output of AGC units. Thus, in this paper, wind turbine is operated in deloading state to provide sufficient reserve capacity. When the wind power reserve capacity $P_{wind}$ is greater than the reference to wind farm power $P_{ref}$, the objective function in wind farm can be designed as

$$J(k) = \sum_{n=0}^{N_c} \left( \|\overline{x}_i(k+n|k)\|^2 + \|u_i(k+n|k) - P_{ref}\|^2 + \sum_{\substack{i,j=1 \\ i \neq j}}^{M} \|u_j(k+n|k-1)\|^2 \right).$$

(18)

The element $\|\Delta u_i(k+n|k) - P_{ref}\|^2$ will makes the output of the wind power as close as possible to $P_{ref}$.

When the wind power reserve capacity $P_{\text{wind}}$ is less than the reference to wind farm power $P_{\text{ref}}$, the objective function in wind farm can be designed as

$$J(k) = \sum_{n=0}^{N_c} \left( \|\bar{x}_i(k+n|k)\|^2 + \|u_i(k+n|k) - P_{wind}\|^2 + \sum_{\substack{i,j=1 \\ i \neq j}}^{M} \|u_j(k+n|k-1)\|^2 \right). \tag{19}$$

The element $\|\Delta u_i(k+n|k) - P_w\|^2$ will make the output of the wind power as close as possible to $P_{\text{wind}}$.

Furthermore, when considering the participation of the wind power, the objective function of thermal power can be designed as

$$J(k) = \sum_{n=0}^{N_c} \left( \|\bar{x}_i(k+n|k)\|^2 + \|u_i(k+n|k)\|^2 + \|(P_{wind} - P_{ref}) - P_{gi}\|^2 + \sum_{\substack{i,j=1 \\ i \neq j}}^{M} \|u_j(k+n|k-1)\|^2 \right). \tag{20}$$

The element $\|(P_{\text{wind}} - P_{\text{ref}}) - P_{gi}\|^2$ indicates the power which needs the thermal power plant to provide to supplement the shortage after the wind power takes part in.

When wind power does not take part in the AGC, the objective function of the thermal power plant can be written as

$$J(k) = \sum_{n=0}^{N_c} \left( \|\bar{x}_i(k+n|k)\|^2 + \|\Delta u_i(k+n|k)\|^2 + \sum_{\substack{i,j=1 \\ i \neq j}}^{M} \|\Delta u_j(k+n|k-1)\|^2 \right). \tag{21}$$

*3.3. Predictive Optimization*

Depending on Formula (17), the state and output variable sequence of the future time series can be calculated as (22) and (23) respectively.

$$X(k) = F_x \cdot x(k) + G_x \cdot U(k), \tag{22}$$

where $N_{\text{p}}$ is the predictive horizon, $N_{\text{c}}$ is the control horizon, $k$ is the sampling instant,

$$X(k) = \begin{bmatrix} x(k+1) \\ \vdots \\ x(k+N_p) \end{bmatrix}, \ U(k) = \begin{bmatrix} u(k+1) \\ \vdots \\ u(k+N_c-1) \end{bmatrix}, \ F_x = \begin{bmatrix} A \\ \vdots \\ A^{N_p} \end{bmatrix}, \ G_x = \begin{bmatrix} B & 0 & 0 \\ \vdots & \vdots & 0 \\ A^{N_c-1} & \cdots & B \\ \vdots & \cdots & \vdots \\ A^{N_p-1} & \cdots & \sum_{i=0}^{N_p-N_c} A^i B \end{bmatrix}.,$$

$$Y(k) = F_y \cdot x(k) + G_y \cdot U(k), \tag{23}$$

where $Y(k) = \begin{bmatrix} y(k+1) \\ \vdots \\ y(k+N_{\mathrm{p}}) \end{bmatrix}$, $F_y = \begin{bmatrix} C^T A \\ \vdots \\ C^T A^{N_{\mathrm{p}}} \end{bmatrix}$, $G_x = \begin{bmatrix} C^T B & 0 & 0 \\ \vdots & \vdots & 0 \\ C^T A^{N_c-1} & \cdots & C^T B \\ \vdots & \cdots & \vdots \\ C^T A^{N_{\mathrm{p}}-1} & \cdots & \sum\limits_{i=0}^{N_{\mathrm{p}}-N_c} C^T A^i B \end{bmatrix}$.

The predictive control process is achieved by solving the object function such as (18)–(21) in sampling instance $k$:

Therefore, at time $k$, the control law is

$$U(k) = -\left(G^{\mathrm{T}} Q G + R\right)^{-1} G^{\mathrm{T}} Q F x(k) \tag{24}$$

where $u(k) = -k_x^T x(k)$, $k_x^T = (1 \ 0 \ \cdots \ 0)\left(G_x{}^{\mathrm{T}} Q_x G_x + R_x\right)^{-1} G_x{}^{\mathrm{T}} Q_x F_x$.

Considering the optimization of the system output, the output of the system needs to be as close as possible to the expected value $W(k+i)$, $i = 1,2,3 \dots N_{\mathrm{p}}$. Therefore, the object function of the output can be written as

$$\min_{U(k)} J_y(k) = \|W(k) - Y(k)\|_{Q_y}^2 + \|U(k)\|_{R_y}^2, \tag{25}$$

where $Q_y$ and $R_y$ are the positive definite and symmetric weighting matrices, $W(k) = [w(k+1) \cdots w(k+N_{\mathrm{p}})]$.

Then, the control law is determined by

$$U(k) = -\left(G_y^{\mathrm{T}} Q_y G_y + R_y\right)^{-1} G_y^{\mathrm{T}} Q_y (W(k) - F_y x(k)) \tag{26}$$

The first element of the sequence is output as the control signal which is given by

$$u(k) = d_y^T (W(k) - F_y x(k)), \tag{27}$$

where $d_x^T = (1 \ 0 \ \cdots \ 0)\left(G_y{}^{\mathrm{T}} Q_y G_y + R_y\right)^{-1} G_y{}^{\mathrm{T}} Q_y$.

The algorithm of the proposed PO-2-DOF-PID control strategy is described as follow:

Step1: Initialize variables $u_i(k)$, $x_i(k)$, $Q$, $R$ ($i = 1,2,3, \dots , M$), obtain the optimal parameters of 2-DOF-PID $K_{\mathrm{p}}$, $K_{\mathrm{i}}$, $K_{\mathrm{d}}$, $P_{\mathrm{w}}$, $D_{\mathrm{w}}$ by Particle Swarm Optimization (PSO) algorithm.

Step2: At $k$ sampling instant, transmit $x_i(k \,|\, k + N_{\mathrm{p}})$ to each interconnected control area $i, j = 1,2,3 \dots M$ and $j \neq i$.

Step3: In area1, compare the relationship of $P_{\mathrm{wind}}$ and $P_{\mathrm{ref}}$, if $P_{\mathrm{wind}} >= P_{\mathrm{ref}}$, then $u_{i\mathrm{w}} = P_{\mathrm{ref}}$, $u_{i\mathrm{t}} = 0$, otherwise $u_{i\mathrm{w}} = P_{\mathrm{wind}}$, $u_{i\mathrm{t}} = P_{\mathrm{ref}} - P_{\mathrm{wind}}$. In area2 and 3, $u_{i\mathrm{t}} = P_{\mathrm{ref}}$.

Step4: Solve $U_x(k) = -\left(G_x{}^{\mathrm{T}} Q_x G_x + R_x\right)^{-1} G_x{}^{\mathrm{T}} Q_x F_x x(k)$, $U_y(k) = -\left(G_y^{\mathrm{T}} Q_y G_y + R_y\right)^{-1} G_y^{\mathrm{T}} Q_y (W(k) - F_y x(k))$ in each areas according to (18)–(21).

Step5: Next sampling instant $k + 1$, update the parameters of wind turbine in each predictive model which include 2-DOF-PID controller.

Step6: Transmit $x_i(k + 1)$ to each control area $j = 1, 2, 3 \dots M$ ($j \neq i$), and return to step 3.

## 4. Simulation and Discussion

The effectiveness of the proposed method is tested on Matlab/Simulink (R2017b, MathWorks, Natick, MA, USA). With a simulation model based on Figure 4 established, wind turbines change the output energy that has been obtained by adjusting the blade pitch angle $\theta$. And the mechanical torque $T_{\mathrm{a}}$ of the wind turbines is regulated with blade pitch angle $\theta$. Then, the output of active power $P_{\mathrm{e}}$ is regulated by changing the speed of the generator $w_{\mathrm{g}}$ through the speed controller of wind turbine. Detailed parameters of the researched three-area interconnected power system are shown in Table 2. Area 1 includes thermal power and 300 wind turbine units. The rated power of each wind turbine

generator is 5 MW. The wind penetration is about 40%, and wind power takes part in AGC. Area 2 includes thermal power and 200 wind turbine units. The rated power of each wind turbine generator is also 5 MW. The wind penetration is about 30%, and wind power did not take part in AGC. Area 3 only has thermal power with a total capacity of 2000 MW. The simulation results are compared among three methods: FO-2-DOF-PID (proposed method), 2-DOF-PID, and distributed MPC (DMPC).

**Table 2.** AGC system simulation parameter.

| Parameter | Area1 | Area2 | Area3 | Unit |
|-----------|-------|-------|-------|------|
| $M_i$ | 10.5 | 10 | 12 | kg·m$^2$ |
| $D_i$ | 2.75 | 2.5 | 3 | s |
| $B_i$ | 35 | 30 | 40 | p.u./Hz |
| $R_i$ | 0.028 | 0.03 | 0.027 | Hz/p.u. |
| $T_{ri}$ | 10 | 10 | 8 | s |
| $T_{gi}$ | 0.1 | 0.1 | 0.08 | s |
| $K_{ri}$ | 0.25 | 0.25 | 0.2 | Hz/p.u. |
| $T_{ti}$ | 0.2 | 0.2 | 0.15 | s |
| $T_{ij}$ | 0.868 | 0.867 | 0.866 | p.u. |
| $J_r$ | 867637 | - | - | kg·m$^2$ |
| $J_g$ | 534.116 | - | - | kg·m$^2$ |
| $N_g$ | 97 | - | - | - |
| $K_p$ | 0.019 | - | - | - |
| $K_i$ | 0.008 | - | - | - |

In order to obtain the best control performance, the parameters of 2-DOF-PID controller are optimized by an improved PSO algorithm. The size of the PSO algorithm is setting as 50 particle swarms with 50 generations. The optimized parameters of the controllers are shown in Table 3.

**Table 3.** The parameters of the 2-DOF-PID.

| Controller Algorithms | Parameter | Area1 | Area2 | Area3 |
|-----------------------|-----------|-------|-------|-------|
| 2-DOF-PID | $K_P$ | 0.9637 | 0.9276 | 0.9209 |
| | $K_I$ | 0.8296 | 0.5632 | 0.8731 |
| | $K_D$ | 0.4562 | 0.4862 | 0.5034 |
| | $P_w$ | 2.1068 | 2.6403 | 1.9866 |
| | $D_w$ | 0.8469 | 0.8694 | 1.0134 |
| PO-2-DOF-PID | $K_P$ | 0.9032 | 0.9586 | 0.9835 |
| | $K_I$ | 0.9691 | 0.4846 | 0.8501 |
| | $K_D$ | 0.3543 | 0.3691 | 0.4305 |
| | $P_w$ | 1.8077 | 2.5035 | 2.2156 |
| | $D_w$ | 0.8969 | 0.7862 | 0.9861 |

During the predictive optimal process, the control horizon $N_c$ and the predictive horizon $N_p$ are set as 4 and 16 respectively. The simulation time is set as 600 s and the sampling period is set as 0.03 s. Additionally, the load fluctuation curves are shown in Figure 5. In order to verify the feasibility and robustness of the proposed method under various wind speed, two cases of high wind speed and low wind speed are analyzed. The wind speed curves are shown in Figure 6. Considering the periodic characteristics of the actual adjustment signal, the zero-order holder is used to process the sampling signal in the load curves and wind speed curves, the periods of both curves are 120 s and 60 s respectively.

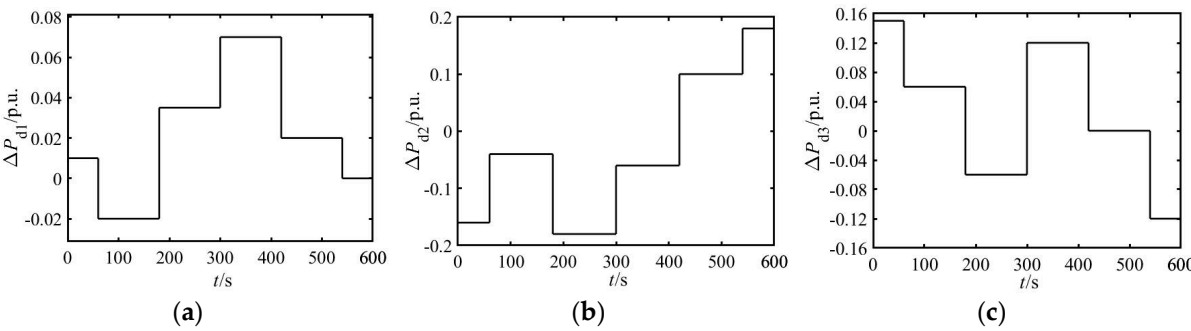

**Figure 5.** Load fluctuation curves in three areas. (**a**) Area1. (**b**) Area2. (**c**) Area3.

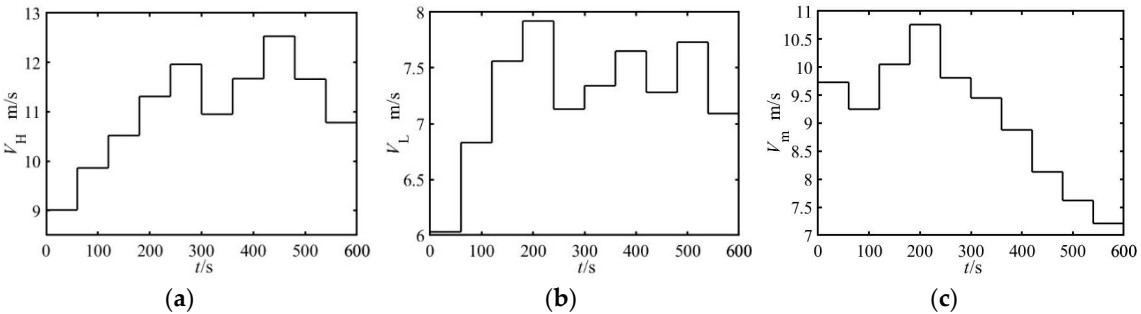

**Figure 6.** Wind speed fluctuation curve. (**a**) High wind speed in area1. (**b**) Low wind speed in area1. (**c**) Wind speed in area2.

**Case 1: Under high wind speed condition**

The control performances of the system frequency in three areas are shown in Figure 7. The *ACE* figures in three areas are shown in Figure 8. The synergetic output of wind power and thermal power for frequency regulation in area 1 is shown in Figure 9, where $\Delta P_d$ is the increase of the load, $\Delta P_e$, $\Delta P_g$ are wind power and thermal power supplementary outputs for the frequency adjustment.

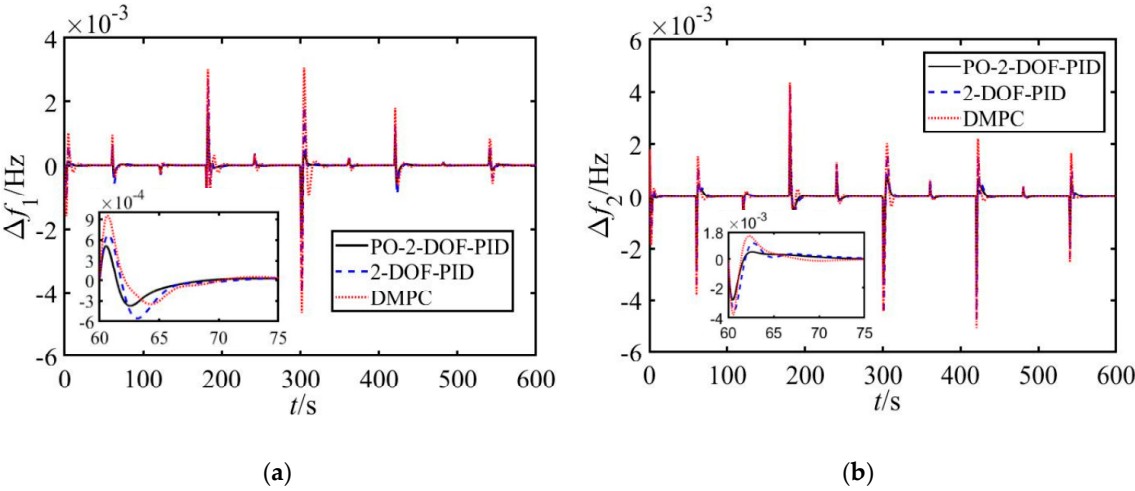

**Figure 7.** *Cont.*

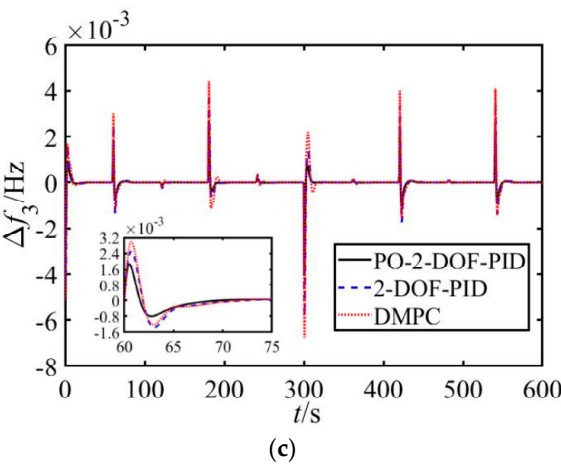

(**c**)

**Figure 7.** Frequency response of power system in high wind speed. (**a**) Area1 (**b**) Area2. (**c**) Area3.

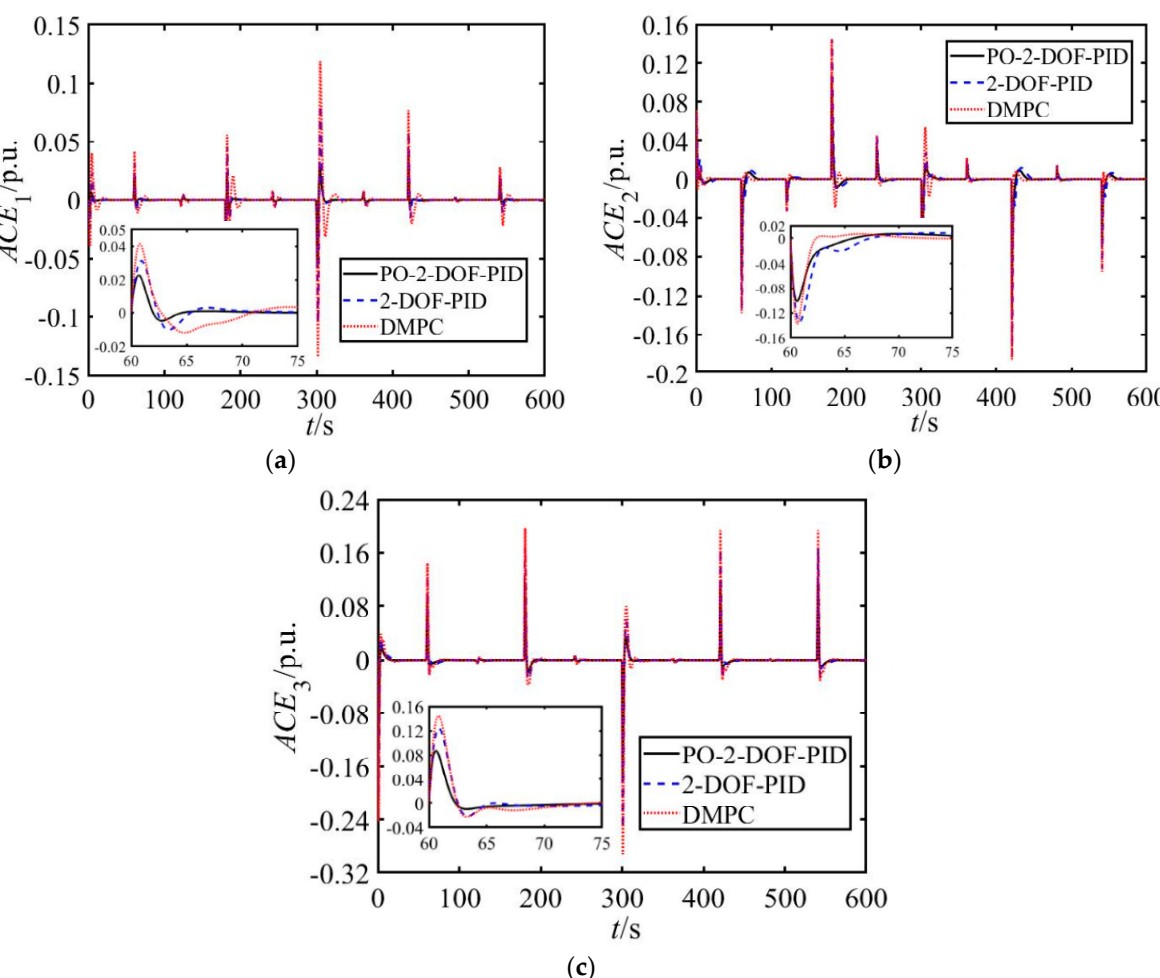

**Figure 8.** ACE of power system in high wind speed. (**a**) Area1 (**b**) Area2. (**c**) Area3.

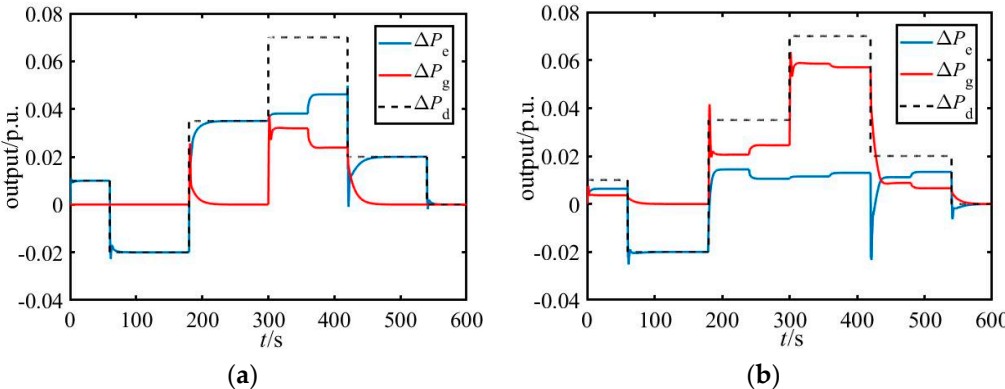

**Figure 9.** Synergetic output of wind power and thermal power for frequency regulation. (**a**) High wind speed. (**b**) Low wind speed.

As shown in Figures 7 and 8, the simulation results indicate that no matter what kind of AGC mode is used in these areas, compared with conventional 2-DOF-PID and DMPC, the proposed PO-2-DOF-PID control method has better control performance in overshoot and setting time etc. When wind power is considered in the AGC, the overshoot caused by the proposed method has at least a 13.63% decrease and a 45.01% decrease compared with 2-DOF-PID and DMPC respectively. In addition, the wind power can supply the power required when load change in most situation under high wind speed, for example, as shown in Figure 9a, during simulation time period 0–300 s and 420–600 s, the wind power can cover the requirement of load change. But during simulation time period 300–420 s, the power capacity which wind turbine provided cannot supply the requirement of the load frequency control. Then, the thermal power units supplement the shortage of the power.

**Case 2: Under low wind speed condition**

Under low wind speed condition, the control performances of the system frequency in three areas are shown in Figure 10. The $ACE_i$ figures in three areas are shown in Figure 11.

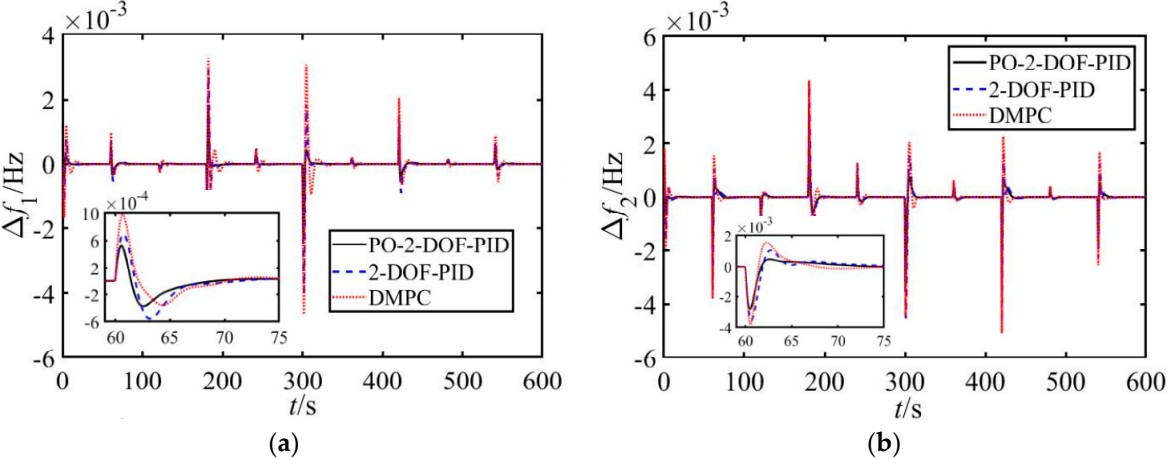

**Figure 10.** *Cont.*

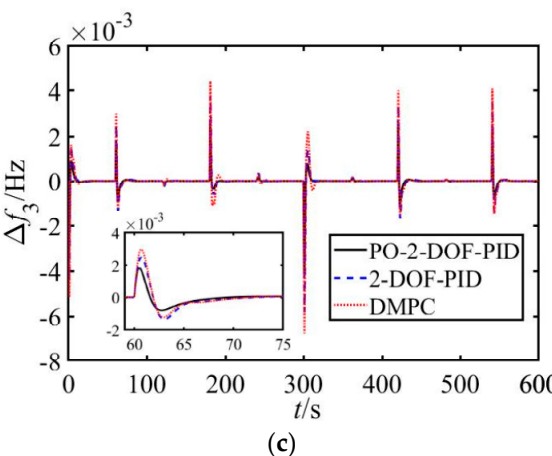

(**c**)

**Figure 10.** Frequency response under low wind speed. (**a**) Area1 (**b**) Area2. (**c**) Area3.

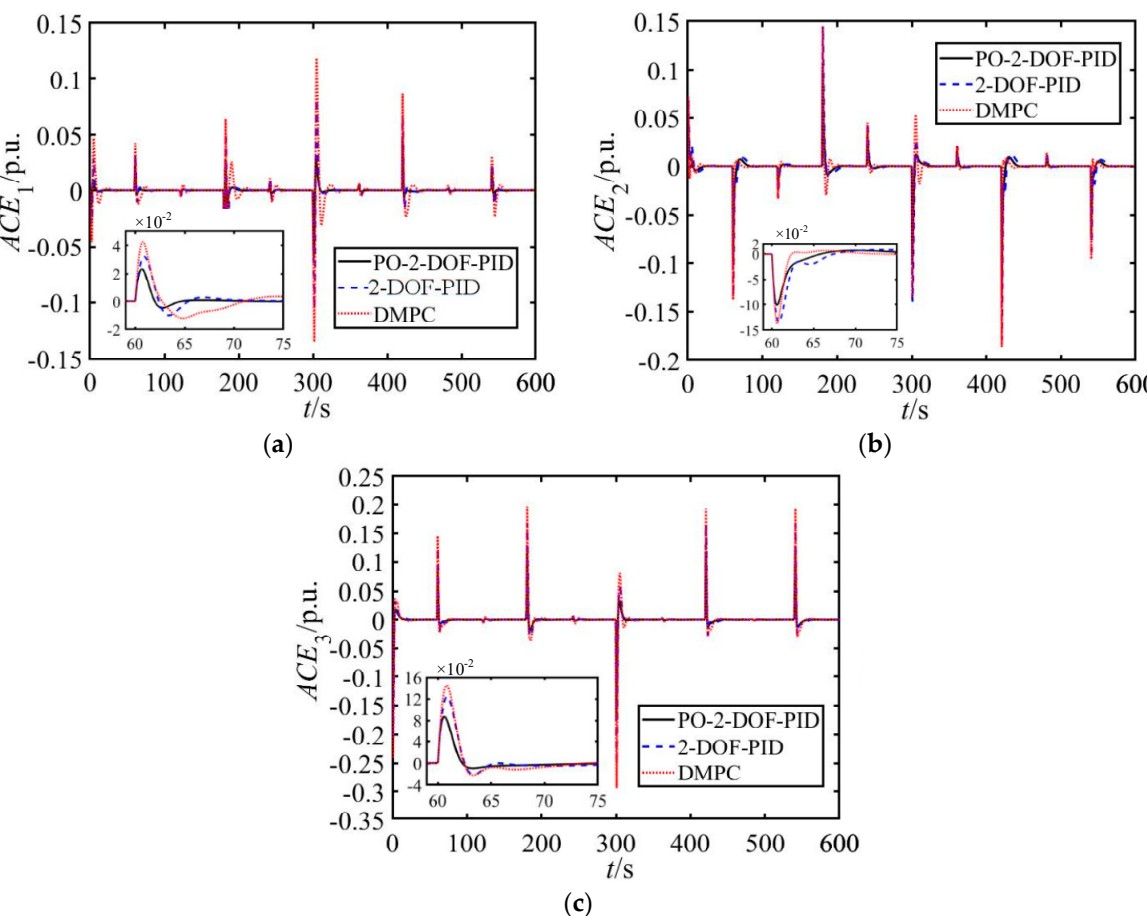

**Figure 11.** ACE of power system under low wind speed. (**a**) Area1 (**b**) Area2. (**c**) Area3.

Similar as case 1, the proposed method has better control performance than conventional 2-DOF-PID and MPC as shown in Figures 10 and 11. Under low wind speed condition, the overshoot caused by the proposed method has at least a 23.23% decrease and 43.44% decrease compared with 2-DOF-PID and DMPC respectively. In addition, although the wind power cannot meet the requirement of the load change, it is the effective complement of traditional AGC which implemented by thermal power, as show in Figure 9b.

## 5. Conclusions

To meet the problem of less reserve power and small inertia constant of the power system when the penetration of wind power is high, to take wind power in the load frequency control is necessary. This paper presents a predictive optimal two-degree-of-freedom PID method for AGC of power system with high penetration of wind power. The main purpose of the design is not only to improve the performance of load frequency control when considering the participation of wind power, but is also to solve the problem about the less flexibility of traditional PID controller which is caused by the fixed parameters. The simulation results show that the wind power can supply the generation required when load change in most situation under high wind speed, and can serve as the effectively complement of traditional AGC under low wind speed. Additionally, compared with conventional 2-DOF-PID and MPC, the proposed method can provide better control performance for the load frequency balance. Based on this conclusion, we hope to introduce the optimized power point tracking of the wind turbine in future in order to use the virtual inertia of the wind turbine to suppress the short-term frequency fluctuation in the power system.

**Author Contributions:** Conceptualization, X.Z. and B.F.; Methodology, X.Z., Z.L.; Data curation, L.H. and N.F.; Writing—Original Draft Preparation, X.Z and Z.L.; Writing—Review and Editing, Z.L. and B.F. All the authors have read and approved the final manuscript.

**Funding:** The research team members thank for the support by the National Natural Science Foundation of China (Grant No. 61473116, No. 51309094), and the Scientific Research Foundation for the Returned Overseas Chinese Scholars, State Education Ministry (Grant No. [2014]1685).

**Acknowledgments:** The authors thank Chaoshun Li for careful reading and helpful suggestions to improve the presentation of this paper.

**Conflicts of Interest:** The authors declare no conflict of interest.

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
