# Peer review of "Research on Automatic Generation Control with Wind Power Participation Based on Predictive Optimal 2-Degree-of-Freedom PID Strategy for Multi-area Interconnected Power System"

_energies, doi:10.3390/en11123325_

Round 1
Reviewer 1 Report
Review of the paper
Title: Research on the AGC Method with Wind Power Participation Based on Predictive Optimal 2-Degree-of-Freedom PID Strategy for Multi-area Interconnected Power System
By: Xilin Zhao, Zhenyu Lin, Bo Fu, Li He, Na Fang
Submitted to: Energies
Manuscript ID: energies-394864
This paper presents a study on the automatic generation control method with wind power participation using predictive optimal 2-Degree-of-Freedom PID strategy for multi-area interconnected power system. The subject is interesting. The scope of the journal totally covers this research topic. I recommend the paper for publication in this journal after revision. Below I have listed the following main issues, which I suggest to take into consideration.
1. In Section 2, list the main assumptions pertaining to the proposed models.
2. In Section 2, define each variable and parameters used in Eqs. 1-15.
3. In Section 3, define each variable and parameters used in Eqs. 16-23.
4. In Section 4, authors should propose tentative interpretation and potential physical reasons behind the described behaviour of their results.
5. In the conclusion, the main outcomes resulting from the study should be clearly identified; suggested improvements and future direction of this work should be emphasised.
6. Run the spelling typing/format mistakes and grammar tool. Given the number of equations, symbols and acronyms used, it may be worthwhile to include a nomenclature section. Avoid acronyms in the title. Improve the quality/presentation of the figures.
Author Response
Dear Reviewer:
We are the authors of manuscript “Research on the AGC Method with Wind Power Participation Based on Predictive Optimal 2-Degree-of-Freedom PID Strategy for Multi-area Interconnected Power System” (ID: energies-394864). We would like to express our thanks to you for your valuable and constructive comments concerning our submission. The manuscript has been revised to fully address the issues raised in the review. In what follows, we present in detail on how the paper has been modified. To enhance readability of the presentation, the comments of you are shown in italics. In the revised paper, all modifications have been marked with red color.
1. In Section 2, list the main assumptions pertaining to the proposed models.
Response:In the revised manuscript, we have added the description of the relevant assumptions in section 2.1 from line 122 to 124. Additionally, we also added a description in section 4 from line 314 to 316. Both complementary descriptions are marked with red color.
2. In Section 2, define each variable and parameters used in Eqs. 1-15.
Response: We carefully examined formula 1-15, and amended a variable identifier respectively in formula 4 and 13 in the revised manuscript. The corresponding formula numbers are marked with red color.
3. In Section 3, define each variable and parameters used in Eqs. 16-23.
Response: We carefully examined formula 16-23, and amended a variable identifier respectively in formula 18 and 20 in the revised manuscript. The corresponding formula numbers are marked with red color.
4. In Section 4, authors should propose tentative interpretation and potential physical reasons behind the described behaviour of their results.
Response:We added a brief description about the mechanism of the output regulation of the wind turbine during the simulation process from line 292 to line 296 in the revised manuscript.
5. In the conclusion, the main outcomes resulting from the study should be clearly identified; suggested improvements and future direction of this work should be emphasised.
Response: We added some quantitative description about the result in section 4. In addition, we also added a description about the future work in section 5. Both are marked with red color.
6. Run the spelling typing/format mistakes and grammar tool. Given the number of equations, symbols and acronyms used, it may be worthwhile to include a nomenclature section. Avoid acronyms in the title. Improve the quality/presentation of the figures.
Response: We examined the spelling and grammar carefully. We deleted some acronyms which only appeared once in the manuscript. And we also modified the acronym in the title “AGC” to its full name “Automatic Generation Control”. Last, we replaced Fig.9 in order to display result clearer.

Reviewer 2 Report
Dear authors.
The submitted manuscript deals with a topic which has considerable practical implications so novel solutions are always welcome.
The paper is well organized, the literature review is relevant to the topic and highlights the main topics in the state of the art and accurately positions the contribution of your work.
The proposed algorithm is an incremental improvement over existing techniques and well supported by the research results.
However, it is important that the equations are reviewed and the nomenclature is checked for inconsistencies: eg
Check the sign in Fig2 for Tij(fi-fj): I think it should be a minus and not a plus
line 142: DPwi is not defined
Pg of Eq 13 is the same as Pe in Fig 3?
line 170: Pref in is the same as in eq 12? if yes, the definitions differ
Author Response
Dear Reviewer:
We are the authors of manuscript “Research on the AGC Method with Wind Power Participation Based on Predictive Optimal 2-Degree-of-Freedom PID Strategy for Multi-area Interconnected Power System” (ID: energies-394864). We would like to express our thanks to you for your valuable and constructive comments concerning our submission. The manuscript has been revised to fully address the issues raised in the review. In what follows, we present in detail on how the paper has been modified. To enhance readability of the presentation, the comments of you are shown in italics. In the revised paper, all modifications have been marked with red color.
1. Check the sign in Fig2 for Tij(fi-fj): I think it should be a minus and not a plus.
Response:Thank you very much for your meticulousness, I am so sorry that we mistakenly wrote Tij(fi-fj) as Tij(fi+fj) in Fig.2. Now, we have made corrections in the revised manuscript.
2. line 142: DPwi is not defined
Response: We define ΔPwi as wind power disturbance, and added this definitions in Table1.
3. Pg of Eq 13 is the same as Pe in Fig 3.
Response: According to the related description in reference [30], The Pg in Eq13 and Pe in Fig.3 are the same physical variables. The Pg is substituted by Pe in formula 13, and we also added a relevant definitions in the revised manuscript.
4. line 170: Pref in is the same as in eq 12? if yes, the definitions differ.
Response:Yes, the Pref in line 170 is the same as in Eq12. And we revised the definition of Pref from “power demand” to “wind farm power reference” according to [30] in the revised manuscript.

Reviewer 3 Report
This paper presents a predictive optimal 2-degree-of-freedom PID method for control of load frequency by wind farm. The concept of paper is interesting, in addition the authors claimed that the proposed method can effectively deal with the negative influence caused by wind power when wind power participates in AGC. However, the method is not well motivated and its advantage is not well explained in the current form. The reviewer’s concerns are listed as follows:
1. In the introduction the authors should clarify the contribution of their proposal compared to the literature they cited. If in the current literature there are issues that can be overcome by the proposed approach, they should be clarified.
2. It is suggested to avoid a wide use of acronyms that make difficult to read the paper.
3. Descriptions and explanations in the text should be improved. For example, how Fig. 2 is derived is unclear and completely left to the reader, which does not ease reading.
4. The authors did not provide a clear overview of the progress beyond the state of the art that is achieved by the proposed methodology. Some example are: Some example are: Reserve Allocation of Photovoltaic Systems to Improve Frequency Stability in Hybrid Power Systems (Energies 11 (10), 2583), Load-frequency control in a multi-source power system connected to wind farms through multi terminal HVDC systems (Computers & Operations Research 96, 305-315). Please include these papers in your references list, and discuss in details in literature review.
Author Response
Dear Reviewer:
We are the authors of manuscript “Research on the AGC Method with Wind Power Participation Based on Predictive Optimal 2-Degree-of-Freedom PID Strategy for Multi-area Interconnected Power System” (ID: energies-394864). We would like to express our thanks to you for your valuable and constructive comments concerning our submission. The manuscript has been revised to fully address the issues raised in the review. In what follows, we present in detail on how the paper has been modified. To enhance readability of the presentation, the comments of you are shown in italics. In the revised paper, all modifications have been marked with red color.
1. In the introduction the authors should clarify the contribution of their proposal compared to the literature they cited. If in the current literature there are issues that can be overcome by the proposed approach, they should be clarified.
Response: In the revised manuscript, we have made a revision in the introduction to illustrate the advantages of our proposal. Firstly, we divided the fourth paragraph into two parts. The first part focuses on the analysis about frequency modulation using the kinetic energy stored in the wind turbine. In the second part, we focus on the analysis about capacity reserve control and pitch angle adjustment. Meanwhile, we improved our analysis in sixth paragraph of the introduction. Furthermore, we made a quantitative analysis on the efficiency improvement of our proposal in the section 4 and conclusion. All the revisions are marked with red color.
2. It is suggested to avoid a wide use of acronyms that make difficult to read the paper.
Response: We deleted some acronyms which only appeared once time in the manuscript. And we also modified the acronym in the title “AGC” to its full name “Automatic Generation Control”.
3. Descriptions and explanations in the text should be improved. For example, how Fig. 2 is derived is unclear and completely left to the reader, which does not ease reading.
Response: We gave a brief description about Fig.2, and added a reference for more details. At the same time, we also modified some relevant identification in Fig.2.
4. The authors did not provide a clear overview of the progress beyond the state of the art that is achieved by the proposed methodology. Some example are: Some example are: Reserve Allocation of Photovoltaic Systems to Improve Frequency Stability in Hybrid Power Systems (Energies 11 (10), 2583), Load-frequency control in a multi-source power system connected to wind farms through multi terminal HVDC systems (Computers & Operations Research 96, 305-315). Please include these papers in your references list, and discuss in details in literature review..
Response:We have revised the introduction as described in comment 1. Additionally, we added two extra references ([3] and [5] in revised manuscript) to make the introduction more complete.

Round 2
Reviewer 1 Report
Authors addressed my comments in the revised version of the paper. The paper can now be accepted for publication in Energies Journal.
Reviewer 2 Report
Dear authors,
Thank you for your reply.
The paper is now suitable for publication.
Reviewer 3 Report
Accept in present form!